

**Characterizing the spatial variations and correlations of**
**large rainstorms for landslide study**
**Liang Gao[1], Limin Zhang[1] and Mengqian Lu[1]**
[1]Department of Civil and Environmental Engineering, The Hong Kong University of Science
and Technology, Clear Water Bay, Hong Kong
*Correspondence to:* L. M. Zhang (cezhangl@ust.hk)
**Abstract**
Rainfall is the primary trigger of landslides in Hong Kong; hence rainstorm spatial distribution
is an important piece of information in landslide hazard analysis. The primary objective of this
paper is to quantify spatial correlation characteristics of three landslide-triggering large storms
in Hong Kong. The spatial maximum rolling rainfall is represented by a rotated ellipsoid trend
surface and a random field of residuals. The maximum rolling 4-h, 12-h, 24-h and 36-h rainfall
amounts of these storms are assessed via surface trend fitting, and the spatial correlation of the
detrended residuals is determined through studying the scales of fluctuation along eight
directions. The principal directions of the surface trend are between 19° and 43°, and the major
and minor axis lengths are 83-386 km and 55-79 km, respectively. The scales of fluctuation of
the residuals are found between 5 km and 30 km. The spatial distribution parameters for the
three large rainstorms are found to be similar to those for four ordinary rainfall events. The
proposed rainfall spatial distribution model and parameters help define the impact area, rainfall
intensity and local topographic effects for landslide hazard evaluation in the future.



## 1    Introduction

Severe rainstorms are one of the most dangerous meteorological phenomena which pose risks
to human lives and properties. A large rainstorm may cause serious damage to infrastructures
and public safety. For instance, a large storm hit Lantau Island, Hong Kong, on 5-7 June 2008
and caused about 2,400 natural terrain landslides and 622 flooding spots (CEDD, 2009). In
hazards mitigation and engineering design, certain 'design storms' must be considered and the
engineering system should be sufficiently safe under such design storms (Gao et al., 2015). A
design storm is often defined by a hyetograph (time distribution) and an isohyet (spatial
distribution). For a particular region where the spatial rainfall variation is significant, a uniform
representation of the spatial distribution is not reasonable since a storm has a centre and
influences a limited area (AECOM and Lin, 2015). Instead, relevant spatial variation factors of
rainfall must be characterized, such as the geometry of spatial form (agglomerate and local
gradient) and the spatial correlation.

A storm is difficult to model due to its intermittence (i.e. no rainfall at a particular position

during a particular short period) and strong spatial and temporal heterogeneity (e.g.,
Barancourt et al., 1992; Bacchi and Kottegoda, 1995; Mascaro, 2013). However, the rainfall
amount, which is in form of regionalized variables, is spatially correlated over a certain
distance (Panthou et al., 2014; de Luca, 2014). A regionalized variable is any variable
distributed in space. Random field theory is recognized as a suitable theory for describing
regionalized variables (Vanmarcke, 1977) and has been proven effective for the regionalized
variables (e.g., Dasaka and Zhang, 2012; Li et al., 2015). The random field theory has also been
used in spatial storm analysis (e.g., Rodríguez-Iturbe, 1984; Bouvier, 2003), and adopted to
describe storm spatial structures (e.g., Zawadzki, 1973; Lebel et al., 1987; Gyasi-Agyei and
Pegram, 2014).

Research on spatial rainfall distribution using statistical models has been performed in





Hong Kong for different engineering purposes (Leung and Law, 2002; Jiang and Tung, 2014;
AECOM and Lin, 2015). Leung and Law (2002) conducted kriging analysis on Hong Kong
hourly rainfall data in 1997 and 1998. Rainfall contours were interpolated to qualitatively
estimate possible flooding locations. Jiang and Tung (2014) derived rainfall
depth-duration-frequency relations at ungauged sites in Hong Kong using an ordinary kriging
approach based on annual maximum daily rainfall data. The extreme rainfall estimates are
sensitive to assumed statistical parameters and uncertainties of the interpolation method.
The storm characteristics such as distribution form and spatial correlation are not
sufficiently analysed when studying the hydrological response of a target system such as a
slope safety system. In particular, limited attention has been paid to event-based spatial
characteristics of large rainstorms in Hong Kong, whose patterns and structures are as useful as
the statistical trend based on historic rainfall records, especially when one needs to select large
rainstorms for landslide risk assessment. Sufficient information should be provided including
both spatial variation and correlation. However, several key questions have not been answered.
Can the spatial precipitation distribution of a large storm be represented using a particular
spatial form? How does the spatial correlation of rainfall change with the rainstorm magnitude?
What are the key factors that influence the spatial structure of rainfall distribution? Such
questions motivate the present study on the spatial characteristics of large rainstorms over hilly
terrains in Hong Kong.
The objective of this paper is to identify the spatial variations and correlation of large
rainstorms in Hong Kong. Three large storms that caused the most severe landslide hazards in
Hong Kong in the past 20 years are selected for study. These storms were often referred to in
Hong Kong as reference storms in preparing engineering measures for landslide hazard
mitigation. The results are therefore expected to provide valuable information for landslide
hazard analysis and risk management.






## 2   Topography and general rainfall distribution in Hong Kong


Hong Kong is located at the southeast coast of China. The subtropical climate in Hong Kong is
characterized by notable dry and wet seasons. About 85% of the annual rainfall is recorded
during the wet season from April to September. Storms with high intensity and short duration
in Hong Kong are typically associated with southwest monsoon or tropical cyclones. The
ground surface elevation on the GIS platform is shown in Fig. 1. The two highest mountain
peaks in Hong Kong are Tai Mo Shan (Near rain gauge N14) and Lantau Peak (Near rain gauge
N21), with peak elevations of 957 m and 934 m above the sea level, respectively. Both the
moisture movements and the topography determine the distribution characteristics (e.g.,
agglomerate and local gradient) of rainfall in the spatial domain.
AECOM and Lin (2015) studied the orographic factors of rainfall spatial distribution
based on historical records. A spatial distribution of orographic intensification factors has been
developed based on historical hourly data. The 24-h orographic intensification factors at a
resolution of 5 km×5 km are shown in Fig. 2. The factors for the land area are in general larger
than those for the sea area. The higher the elevation is, the larger the orographic intensification
factor. Two of the highest intensity regions are located at Tai Mo Shan in New Territories and
Lantau Peak on Lantau Island. The trend of the factors coincides with the mountain range
alignment, i.e., around N45°E.
The magnitude of storms can be assessed corresponding to a depth-area relation, and
characterized by the probable maximum precipitation (PMP). PMP is frequently used to
quantify extreme storm events (WMO, 2009). The scenarios of 4-hour and 24-hour PMP for
Hong Kong have been assessed by Hong Kong Observatory and AECOM (Chang and Hui,
2001; AECOM and Lin, 2015). AECOM and Lin (2015) updated the 24-h PMP for Hong Kong
considering the local orographic intensification. The trend surface is an expected-value





surface. The trend surfaces of 24-h PMP with different storm centres have been updated by
AECOM and Lin (2015), and the typical trends are shown in Fig. 3. The trend surfaces are
derived based on the historical hourly rainfall. According to the 24-h PMP updating study, an
elliptical isohyet is recommended as a generalized convergence pattern. For storms cantered at
Tai Mo Shan, the orientation of 22.5° (N 67.5° E) is found to be the most critical.

**3  Progression and precipitation data of three large storms**
The most traditional way to describe the rainstorm severity is by return period, which is
recognized as a combination of intensity and duration. Another measure of the severity of a
storm is the consequence of the storm, such as rain-induced landslides or flooding. An index
measuring the potential to trigger landslides, named "Landslide Potential Index (LPI)", is used
in Hong Kong (CEDD, 2009). The LPI is based on the historic records of landslide events since
1984. For instance, a storm in late July 1994 caused 5 fatalities and its LPI was 10. The value of
LPI can be greater than 10 if a storm is more damaging than the July 1994 storm. According to
the LPI, the top three largest storms in the past 20 years were the 5-7 June 2008 storm, the
17-21 August 2005 storm and the 23 July 1994 storm. Each of these three storms had an LPI
around 10 and led to fatalities. Thus, the three storm events are selected as indicative large
storms to conduct spatial correlation analysis in this paper.
The rainfall data in this study is provided by Geotechnical Engineering Office (GEO) and
Hong Kong Observatory (HKO) in Hong Kong. The GEO and HKO rain gauge networks
comprise 88 and 46 stations, respectively (Fig. 1). The rain gauges are more concentrated in the
northern Hong Kong Island and Kowloon where the population density is high. The raw digital
data at 5-minute interval from the high quality network ensures the reliability of this study. The
data covers the period from 00:00 on 5 June to 24:00 on 7 June 2008, from 00:00 on 17 August
to 24:00 on 21 August 2005, and from 00:00 on 22 July to 24:00 on 24 July 1994. Some of the



rain gauges had not been installed in July 1994. The numbers of effective rain gauges for the
three events are 105, 112, and 56, respectively. The three storm hyetographs corresponding to
the maximum local precipitation depth are shown in Fig. 4. The 17-21 August 2005 storm is
more moderate in short durations compared with the 5-7 August 2008 storm and the 22-24 July
1994 storm.

**3.1 The 5-7 June 2008 storm**
According to Hong Kong Observatory (HKO), the weather was influenced by an active low
pressure trough over the south China coastal area during the first 10 days of June 2008, and was
heavily rainy and thundery. Fig. 5 (a) shows contours of the total rainfall amount of the 5-7
June 2008 storm. The maximum total rainfall amount was 670 mm. The storm centre was on
the southeast of Lantau Island. The magnitudes of the storm characterized by 4-h PMP and
24-h PMP (AECOM and Lin, 2015) are shown in Fig. 6. From the depth-area relationships,
when the area is in the range of 50 - 1100 km$^2$, the maximum rolling 4-h rainfall of the 5-7 June
2008 storm has a return period of 1,100 years, corresponding to 60%-67% of the 4-h PMP,
while the return period for the 24-h rainfall is 200 years, corresponding to 33%-41% of the
24-h PMP. The storm caused 2,400 natural terrain landslides (Li et al., 2009), including many
debris flows that affected developed regions, leading to 2 fatalities (CEDD, 2008). The LPI
value was recognized as 12. The 4-h maximum rolling rainfall value is calculated as the
maximum values of rainfall in 4 consecutive hours on a hyetograph.

The maximum rolling rainfall values at different locations may not be in the same period

though most of them tend to be in the same period. Hazard consequences are more related to
the maximum rolling rainfall values other than instantaneous one (Dai and Lee, 2001). In
formulations for a hydrological model, the effect of the time scale of aggregation of the rainfall
data and the hydrological response of catchments of different sizes should be investigated in



order to identify the critical scale at which the resulting discharge will be the largest and could
potentially generate flash floods.

The most concentrating periods of precipitation are selected. Figure 7 shows the

instantaneous rainfall process from 6: 55 to 7: 35 on 7 June 2008. During this period, the
vapour concentrated on the southwest of Lantau Island, and transported northeast across the
mountains on Lantau Island. A large amount of precipitation was retained on the island.

**3.2 The 17-22 August 2005 storm**
August 2005 was much wetter than normal. A very active southwest monsoon during 17-22
August brought in plenty of moisture. Figure 3(d) shows contours of the total amount of
rainfall. The maximum total rainfall amount was 890 mm. The storm centre was at the middle
of the territory, Shatin. From Fig. 4, both the maximum rolling 4-h rainfall and 24-h rainfall of
the 17-22 August 2005 storm are least critical among the three storms investigated in this
paper. The storm caused 229 reported landslides, resulting in one fatality. The LPI value is 10
(Kong and Ng, 2006).

Figure 8 shows the instantaneous rainfall process from 10:35 to 11:15 on 20 August, 2005,

which is recognized as the heaviest rainfall period in this storm event. The prevailing moisture
inflow mainly came southerly during this period. The rainfall centre concentrated on the south
of Tai Mo Shan.

**3.3 The 21-24 July 1994 storm**
The total precipitation amount in the storm event from 21 to 24 July 1994 was recorded as the
highest for any consecutive days in July. The weather was related to a trough of low pressure
(Tam et al., 1995). Figure 3 shows contours of the total amount of rainfall of this storm
cantering at the middle of New Territories, Tai Mo Shan. The maximum total rainfall amount



was 1450 mm. In Fig. 4, the maximum rolling 24-h rainfall is the most critical, especially for a
smaller area. The storm caused 820 natural terrain landslides and 451 man-made slope failures,
resulting in 5 fatalities and 4 injuries. The LPI value is 10 (Chan, 1995).
Figure 9 shows the instantaneous rainfall process from 15:00 to 15:40 on 23 July 1994,
which records the heaviest rainfall process in this storm event. During this period, the moisture
air came from on the northwest of Tai Mo Shan. Most of precipitation concentrated on Tai Mo
Shan, and the spatial distribution of rainfall was quite uneven. As the moisture flux rose across
Tai Mo Shan, a large amount of moisture began to fall as rain. The orographic intensification
effect was very significant in this rainstorm event.

**3.4 Summary of the three large storms**
All the aforementioned three storms are related to monsoons other than typhoons. The
meteorological factors for these storms are beyond the scope of this paper. This research
focuses on the areal distribution of precipitation which is believed to be more relevant to the
evaluation of the performance of the slope safety system. Thus the maximum rolling rainfall
values are estimated in different durations. According to the records from the automatic rain
gauges, the maximum rolling rainfall among all the rain-gauge stations in each of the three
events can be calculated. The corresponding peak values and stations are summarized in Table
1. The 22-24 July 1994 storm is the largest among the three storms with regard to the amounts
of the maximum rolling 1-h and 24-h rainfall. However, in terms of the maximum rolling 4-h
rainfall, the 5-7 June 2008 storm is the most critical.
The contours of the total rainfall for the three storms, interpolated using a triangular
method, are shown in Fig. 5. The total precipitation amount of the 5-7 June 2008 storm is the
smallest among the three events, while that of the 21-24 July 1994 storm is the largest due to its
longer duration. However, the LPI value for the 5-7 June 2008 storm is 12, larger than those of





the other two storms; that is, the 5-7 June 2008 storm is the largest one in terms of damage. One
of the reasons is that the variability of spatial and temporal distributions of the storm affects
both the infiltration dynamics of the surface soil and the water levels above and below the
ground surface. The entire hydrological system is governed by the spatial and temporal
distribution of rainfall.

**4    Methodology of spatial analysis**
The varying space-time distribution of rainfall in Hong Kong is a result of the interaction
between governing meteorological covariates and local hilly terrain. Instead of attempting the
use of a physical model to capture the spatial characteristics, our analysis presents a two-step
approach in which a surface trend is firstly established to assess the spatial distribution of the
rainfall amount in a fixed duration, followed by a further analysis of the spatial correlation of
the detrended residuals.

**4.1 Determination of the expected precipitation trend surface**
A storm is a phenomenon with gradual geographical changes in space; the rainfall amount can
be simulated as a spatially correlated random field superimposed on a trend surface (Grimes
and Pardo-Igúzquiza, 2010). Such an artificial rainfall trend surface can be used to represent
design storms. One could comprehend that the rainfall is correlated with the local terrain and
the design storm centres are likely to be around the mountain peaks. Hong Kong has a
relatively small area, and an individual storm is usually designed to have one or two centres for
engineering design purposes (AECOM and Lin, 2015). Distinguishing two peaks is not
necessary as the distance between any two peaks will be small with regard to the scale of a
typical rainstorm.

Based on random field theory (Vanmarcke, 1977), the trend surface is the expected value





of the precipitation distributed over the rainfall domain, while the residuals are stationary and
not affected by any shift in the coordinate system. Thus the first step is to divide the spatial
distribution into a trend surface and residuals by finding a trend surface fitting function.
Though most natural processes like a storm exhibit spatial variability with complex trends, this
paper uses a polynomial function for simplicity. Denote observations of a storm as $z_i$ ($x_i$, $y_i$)
(i=1, 2, …, n). The fitted values are $\hat{z}_i = (x_i, y_i)$:
$$z_i(x_i, y_i) = \hat{z}_i(x_i, y_i) + \varepsilon_i \qquad (3)$$
where x and y define the location; and $\varepsilon_i$ are residuals. The second-order polynomial trend
surface is:
$$\hat{z}_i = a_0 + a_1 x_i + a_2 y_i + a_3 x_i^2 + a_4 x_i y_i + a_5 y_i^2 \qquad (4)$$
The coefficients, $a_0$, $a_2$,…, $a_5$, are determined by minimizing the sum of the squares of the error
term using the ordinary least squares (OLS) analysis (Journel and Huijbergts, 1978):
$$Q = \min \sum_{i=1}^{n} \varepsilon_i^2 = \min \sum_{i=1}^{n} [z_i(x_i, y_i) - \hat{z}_i(x_i, y_i)]^2 \qquad (5)$$

The computed trend surfaces for the total rainfall amounts of the three storms and the

detrended residuals are shown in Fig. 10. The residuals of the rainfall amounts in different
durations are often assumed to be stationary. Taking the maximum 4-h rolling rainfall as an
example, the trend surface is
$$\hat{z} = -45984 - 0.0337x + 0.1527y + (-1.5297x^2 + 3.4783xy - 2.7125y^2) \times 10^{-7} \qquad (6)$$

The peak point on the surface is (77429, 77793); the maximum 4-h rainfall on the trend

surface is 425 mm. The maximum points (extreme values) on the trend surfaces of the three
storms are summarized in Table 3. The major and minor axes can be calculated as those of the
ellipse with rainfall value approaching zero. The directions and lengths of the trend surfaces are
summarized in Table 4. The major and minor axes of the trend surfaces are determined by least
squares fitting of the original rainfall data. For an individual storm event, the maximum points





of the trend surfaces are inside a relatively small range of 40 km. The storm centre of each
event on the trend surface agrees with the reality. The storm centres of the 7 June 2008 storm,
the 17-21 August 2005 storm and the 23 July 1994 storm are at west Lantau Island, Shatin and
Tai Mo Shan, respectively. The major directions of the spatial forms are between 19° and 43°
in the anticlockwise direction.

**4.2 Determination of the scale of fluctuation of precipitation residuals**
A classical way to characterizing the spatial correlation is through an autocorrelation function
(ACF), $\rho(h)$ (Fenton and Griffiths, 2008; Foresti and Seed, 2014). The autocorrelation
describes the correlation between values of a same series. The autocorrelation r (k) for lags
k=0, 1, …, m, where m is the maximum number of lags, is evaluated by the following equation:
$$r_k = \frac{\dfrac{1}{(N-k-1)}\sum_{i=1}^{N-k}(z_i - \bar{z})(z_{i+k} - \bar{z})}{\dfrac{1}{(N-1)}\sum_{i=1}^{N-k}(z_i - \bar{z})^2} \tag{7}$$

where $z_i$ and $z_{i+k}$ are the detrended storm depths at locations $i$ and $i+k$, respectively; $N$ is the
total number of the residuals; and $\bar{z}$ is the mean value of the residuals.
In order to assess the autocorrelation structure of the detrended storm amounts, it is
necessary to perform regression analysis to fit the ACF. Among many correlation structures,
the single exponential structure is the most common:
$\rho(h) = \exp(-2h/\theta)$        (8)
where $h$ is the separation distance or lag; $\theta$ is the scale of fluctuation (SoF). The correlation $\rho$
($h$) decays exponentially with separation distance $h$. The negative autocorrelation coefficient
will not be evaluated. The values of $\theta$ can be obtained accordingly. Within the scale of
fluctuation, the rainfall property is strongly correlated. A smaller scale of fluctuation indicates
more rapid fluctuations of the mean.



The scale of fluctuation is evaluated in the directions of N 0° E, N 45°E, N 90° E, and N
135° E for each storm. The values of SoF are fitted by an ellipse using least squares fitting. The
values of SoF and the fitting curves are shown in Figs. 11-13. Greater SoF values indicate
smaller variability. The major direction can be recognized as the direction of maximum
continuity.
The direction and major and minor scales of fluctuation are summarized in Table 4. The
SoF values of the rainfall residuals are between 6 to 37 km. Regardless of the variations of the
principal axis direction, the minor-axis lengths of the SoF values remain around 7 km (Table

4).


**5   Spatial description of rainstorms**
**5.1 Geometric spatial form and correlation structure**
Though rainfall varies over space, the rainfall amount of a particular storm in terms of
maximum rolling rainfall can be fitted by a polynomial function. The spatial form of the
rainfall amount can be represented by a rotated ellipsoid with only one centre. Such an artificial
spatial form may exhibit geometrical regularity. For each storm, the trend surfaces in different
durations show good consistency in the shape parameters in terms of the peak point, long-axis
direction and axis length. The peak points on the trend surfaces of the three storms are located
in a relatively small range. The long-axis directions of the spatial forms of each event in
different durations almost remain unchanged between 19° and 43°. The lengths of the major
and minor axes for an individual storm show consistency. The 5-7 June 2008 storm has the
largest impact area, as indicated by larger axis lengths among the three rainstorms according to
the results in Table 3.
With respect to the instantaneous rainfall processes shown in Figs. 7-9, the rainfall
distributions in terms of maximum rolling rainfall are quite consistent to the heaviest rainfall





process in each storm event. The rainfall distributions are strongly affected by the storm
humidity transportation, and are so uneven that the entire area should not be described as a
single site. The locations of the storm centres determine the general trend of the areal rainfall
distribution. The polynomial trend surfaces are effective for representing large rainstorm
distributions in terms of maximum rolling rainfall.

The spatial connectivity can be assessed by the SoF values. A smaller scale of fluctuation

indicates more rapid fluctuations of the mean. According to Figs. 11-13, all of the SoF values
are within 30 km, though the semi-lengths of the major axes of fitting curves are larger. Hence
a reasonable upper threshold for the spatial connectivity is estimated to be 30 km. On the other
hand, the lengths of the minor axis of the SoF values are between 5 to 8 km. The lower limit of
the SoF values of the rainfall data is considered to be 5 km. Therefore, the rainfall amount in
Hong Kong is observed to be strongly spatially correlated within 5 km, whose spatial
continuity is smaller than 30 km.

## 309     5.2 Comparison with the spatial structures of ordinary rainfall events

Besides the three large rainstorm events in this paper, ordinary rainstorm events in Hong Kong
have also been studied (Liu, 2013; AECOM and Lin, 2015). Liu (2013) proposed a framework
for analysing dynamic time-space evolution of rain-field in her thesis. Four rain events were
chosen to illustrate the spatial structure of rainfall in Hong Kong: the 18 May 2007, 19 May
2007, 19 April 2008 and 15 September 2009 rain events in Hong Kong. The 2008-04-19
rainstorm event was under a combined effect of Typhoon Neoguri and a northeast monsoon,
while the other three rainstorms were results of tropical depressions. The total rainfall amounts
during the four rainfall events on 18 May 2007, 19 May 2007, 19 April 2008 and 15 September
2009 were 67.0, 99.6, 157.9 and 130.3 mm, respectively. The spatial structures of the four rain
events indicated by variogram ranges corresponding to the peak rainfall intensity (six minutes



resolution) are plotted in Fig. 14. According to the results from ellipse fitting, the major
principal directions of all the tropical depression storms (i.e. on 18 May 2007, 19 May 2007,
and 15 September 2009) are around 45°. The lengths of the principal axis of the tropical
depression storms are within 30 km; while that of the 19 April 2008 storm is 40.8 km. The
correlation structures of the instantaneous rain processes are consistent with those of the three
large storms as illustrated in Section 5.1.

The spatial structure of annual maximum daily rainfall using the variogram model

provides additional information for generating design storms from another point of view.
According to the study conducted by Jiang and Tung (2014), the spatial variability represented
by a variogram is used to establish the rainfall depth-duration-frequency relationships. By
normalising the indicator semivariogram by the variance of the indicator data, the normalised
semivariances of the mean annual maximum daily rainfall and the maximum rolling 24 hour
rainfall of the three storms are shown in Fig. 15. Based on the samples and the fitted
exponential variogram model, the range of the mean of annual maximum daily rainfall is 7.1
km, which is close to the omnidirectional range values of the maximum rolling 24-hour rainfall
for the storms, particularly those for the 2008 storm and the 2005 storm. Thus, given a large
storm whose spatial distribution is relatively smooth, the range value will be close to that of the
annual maximum daily rainfall. The spatial structures of the three severe storms and the four
ordinary rainfall events do not differ significantly.

With aspect to the local terrain impacts, the major directions of both the three large

rainstorms and the ordinary rainfall events are all consistent with the mountain range alignment
in Hong Kong (Fig. 1). However the severe storms are highly uncertain and it is difficult to
ascertain and predict the future precipitation and extreme rainfall. Lu et al. (2013), Lu and Lall
(2016) and Najibi et al. (2017) suggest a potential direction to further study the associated
atmospheric circulation with moisture transport that has improved the predictability of extreme



rainfall and flood in various regions including western Europe, Midwest and Northeast of the
United States. The spatial structure found in this study also indicates that there might be a link
between the distribution and the convergence of the moist air into the Hong Kong region.

**6   Conclusions**
A random rain field model has been proposed to study the spatial characteristics of three large
landslide-triggering rainstorms in Hong Kong. The cumulative rainfall depths in terms of
maximum rolling rainfall in different durations are of particular importance for landslide
studies, and are taken as random variables in this study. Based on the study, the following
conclusions can be drawn:
(1)  The amounts of maximum rolling rainfall in different durations share a dominating spatial

structure that can be represented by a rotated ellipsoid surface established using the

ordinary least squares method. The shapes change slightly in different durations for a

particular storm.

(2)  The major principal directions of the surface trends of the three rain storms are between

19° (N 71° E) and 43° (N 47° E), and the principal major and minor axis lengths are

83-386 km and 55-79 km, respectively.

(3)  The spatial connectivity of large storms in Hong Kong is estimated to be between 5 km

and 30 km. The rainfall amounts in the three large storms are observed to be strongly

correlated within 5 km and likely to be connected within 30 km.

(4)  To verify the rationality and reliability of the spatial structures of large rainstorms, the

spatial characteristics of four ordinary rainfall events are also studied. The spatial

structures of the three large rainstorms are similar with those of the ordinary rainfall

events and consistent with the mountain range alignment in Hong Kong.




**Acknowledgements**

The authors would like to thank the Geotechnical Engineering Office (GEO) of the Civil
Engineering and Development Department (CEDD) for providing the rainfall data described in
this paper. This research is supported by the Research Grants Council of the Hong Kong SAR
(Nos. C6012-15G and 16202716).

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

Table 1. Values of maximum rolling rainfall of three landslide-triggering storms in Hong Kong.

| Duration | 5-7 June 2008 storm | | 17-21 August 2005 storm | | 22-24 July 1994 storm | |
|---|---|---|---|---|---|---|
| | Amount (mm) | Station | Amount (mm) | Station | Amount (mm) | Station |
| 1-hour | 154 | N21 | 82 | N25 | 212 | N14 |
| 4-hour | 384 | N19 | 174 | N18 | 365 | N14 |
| 24-hour | 623 | N19 | 570 | N01 | 956 | N14 |
| 2-day | 672 | N19 | 768 | N01 | 1216 | N14 |
| 4-day | 768 | N19 | 890 | N01 | 1450 | N14 |

Table 2. Locations of maximum rainfall on the trend surfaces (km).

| Duration | 5-7 June 2008 storm | 17-21 August 2005 storm | 22-24 July 1994 storm |
|---|---|---|---|
| 4-hour | (774, 778) | (822, 816) | (822, 836) |
| 12-hour | (764, 788) | (825, 822) | (822, 835) |
| 24-hour | (781, 752) | (829, 819) | (823, 833) |
| 36-hour | (769, 747) | (830, 820) | (825, 826) |





Table 3. Directions and lengths of the axes of trend surfaces.

| Duration | 5-7 June 2008 storm | | | 17-21 August 2005 storm | | | 22-24 July 1994 storm | | |
|---|---|---|---|---|---|---|---|---|---|
| | Major axis direction (°) | Major axis length (km) | Minor axis length (km) | Major axis direction (°) | Major axis length (km) | Minor axis length (km) | Major axis direction (°) | Major axis length (km) | Minor axis length (km) |
| 4-hour | 36° | 229 | 61 | 42° | 107 | 56 | 19° | 100 | 72 |
| 12-hour | 29° | 253 | 65 | 40° | 97 | 58 | 40° | 87 | 62 |
| 24-hour | 25° | 269 | 71 | 38° | 85 | 55 | 39° | 92 | 77 |
| 36-hour | 27° | 386 | 65 | 35° | 86 | 55 | 43° | 83 | 79 |

Table 4. Directions and semi-lengths of the axes of scale of fluctuation (SoF).

| Duration | 5-7 June 2008 storm | | | 17-21 August 2005 storm | | | 22-24 July 1994 storm | | |
|---|---|---|---|---|---|---|---|---|---|
| | Major axis direction (°) | Semi-lengths of the major axes (km) | Semi-lengths of the minor axes (km) | Major axis direction (°) | Semi-lengths of the major axes (km) | Semi-lengths of the minor axes (km) | Major axis direction (°) | Semi-lengths of the major axes (km) | Semi-lengths of the minor axes (km) |
| 4-hour | -18° | 31 | 9 | -3° | 14 | 5 | 8° | 10 | 7 |
| 12-hour | -7° | 17 | 7 | 38° | 37 | 7 | 21° | 9 | 6 |
| 24-hour | -36° | 12 | 8 | 33° | 23 | 7 | 4° | 9 | 6 |
| 36-hour | -79° | 18 | 6 | 36° | 24 | 7 | 9° | 7 | 6 |





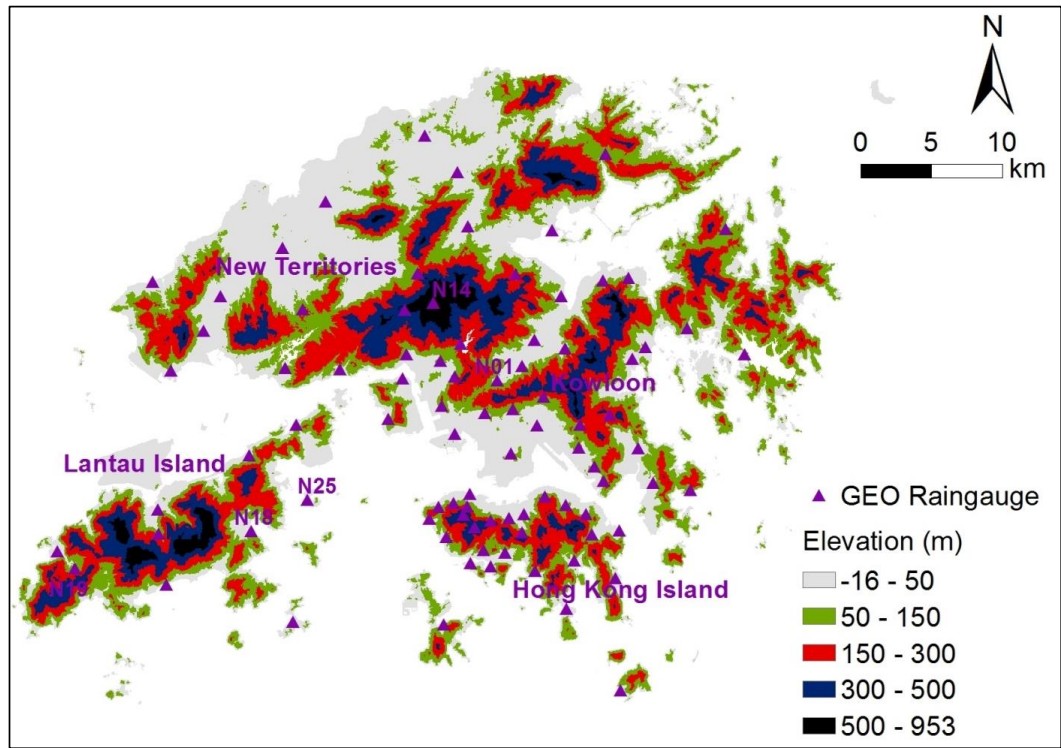

Figure 1. The GEO rain-gauge network in Hong Kong.





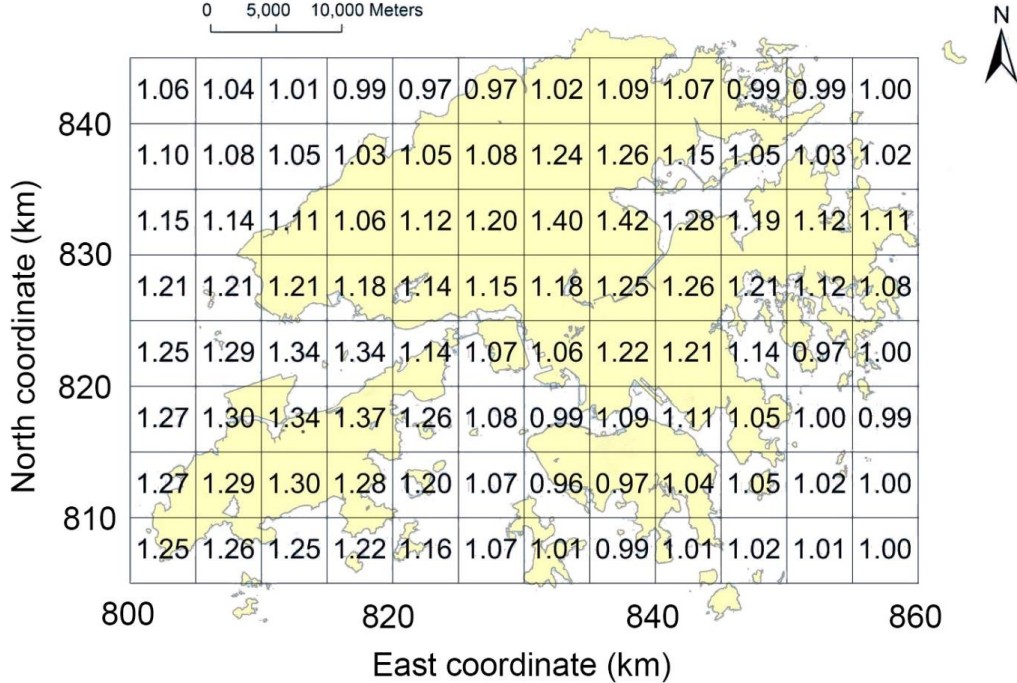

Figure 2. 24-hour orographic intensification factors in Hong Kong (modified from AECOM 2011).



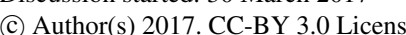


Figure 3. Generalized convergence component pattern with (a) NE-SW orientation 45° (b) ENE-WSW orientation 22.5° centred at Hong Kong Island; (c) NE-SW orientation 45° (d) ENE-WSW orientation 22.5° centred at Lantau Island; (e) NE-SW orientation 45° (f) ENE-WSW orientation 22.5° centred at Tai Mo Shan (modified from AECOM 2011).



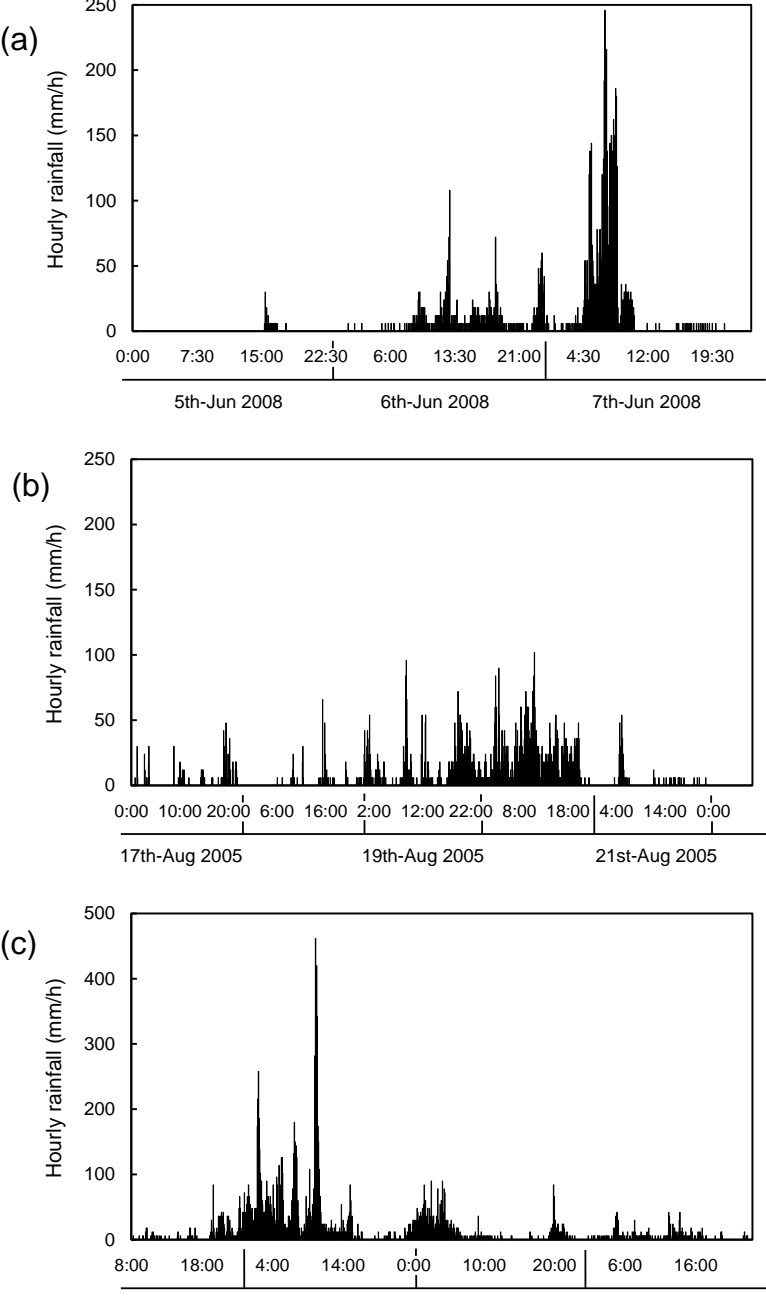

Figure 4. Hyetographs of three storms: (a) 5-7 June 2008 storm, Station N19; (b) 17-21 August 2005 storm, Station N01; (c) 22-24 July 1994 storm, Station N14.



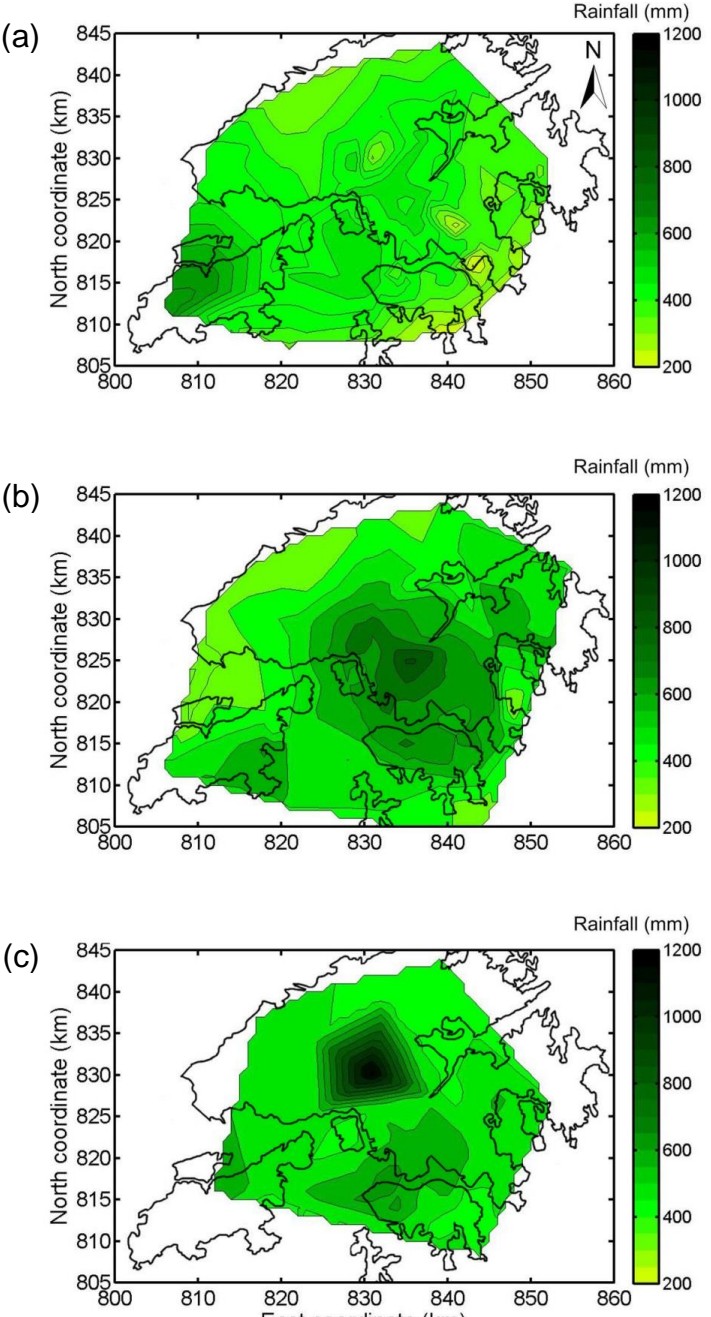

Figure 5. Spatial distribution of the total rainfall amount: (a) the 5-7 June 2008 storm; (b) the 17-21 August 2005 storm; (c) the 22-24 July 1994 storm.





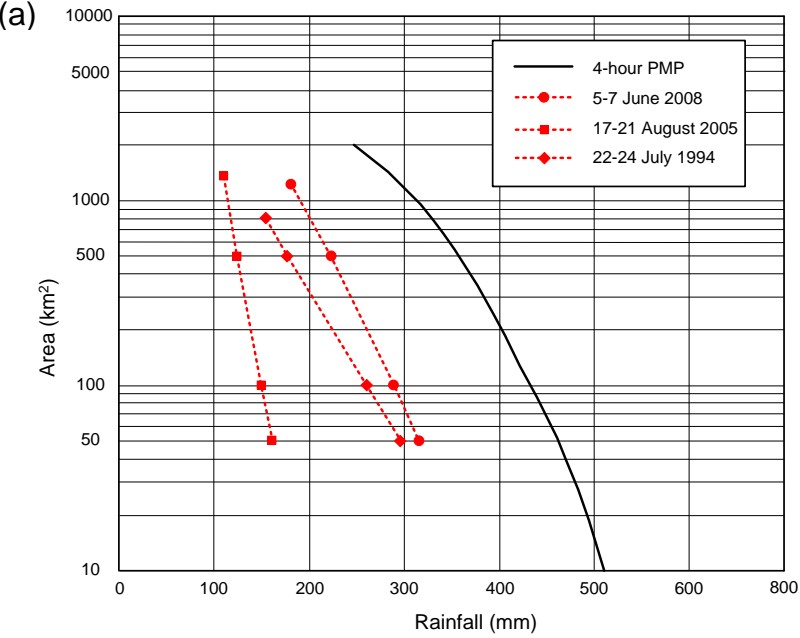

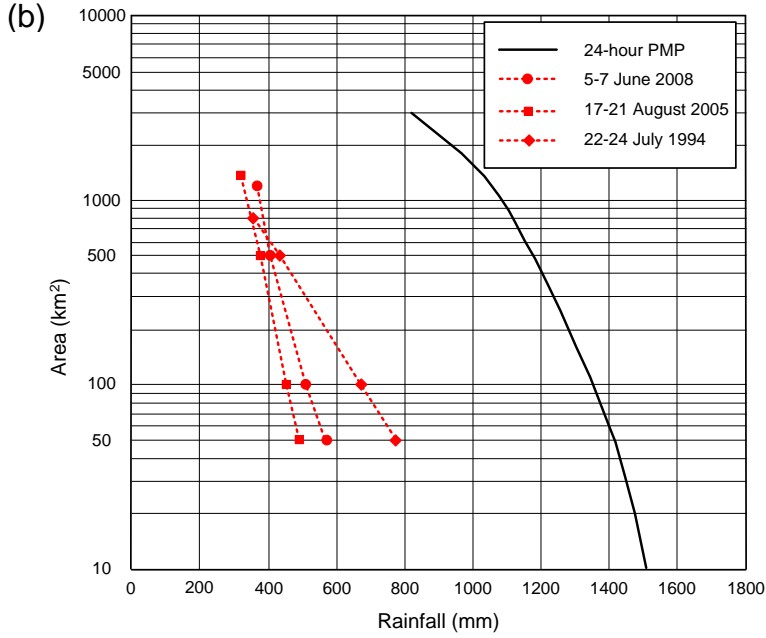

Figure 6. Magnitudes of the three storms characterized by (a) 4-h PMP, and (b) 24-h PMP (Modified from AECOM 2011).





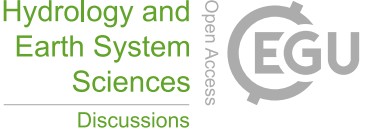

Figure 7. Instantaneous rainfall process from 6:55 to 7:35 on 7 June 2008.

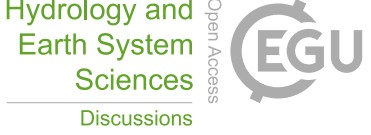

Figure 8. Instantaneous rainfall process from 10:35 to 11:15 on 20 August 2005.

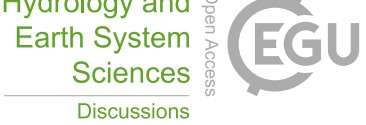

Figure 9. Instantaneous rainfall process from 15:00 to 15:40 on 23 July 1994.





Figure 10. Trend surfaces and residuals of the total rainfall amounts: (a) and (b) the 5-7 June
2008 storm; (c) and (d) the 17-21 August 2005 storm; (e) and (f) the 22-24 July 1994 storm.




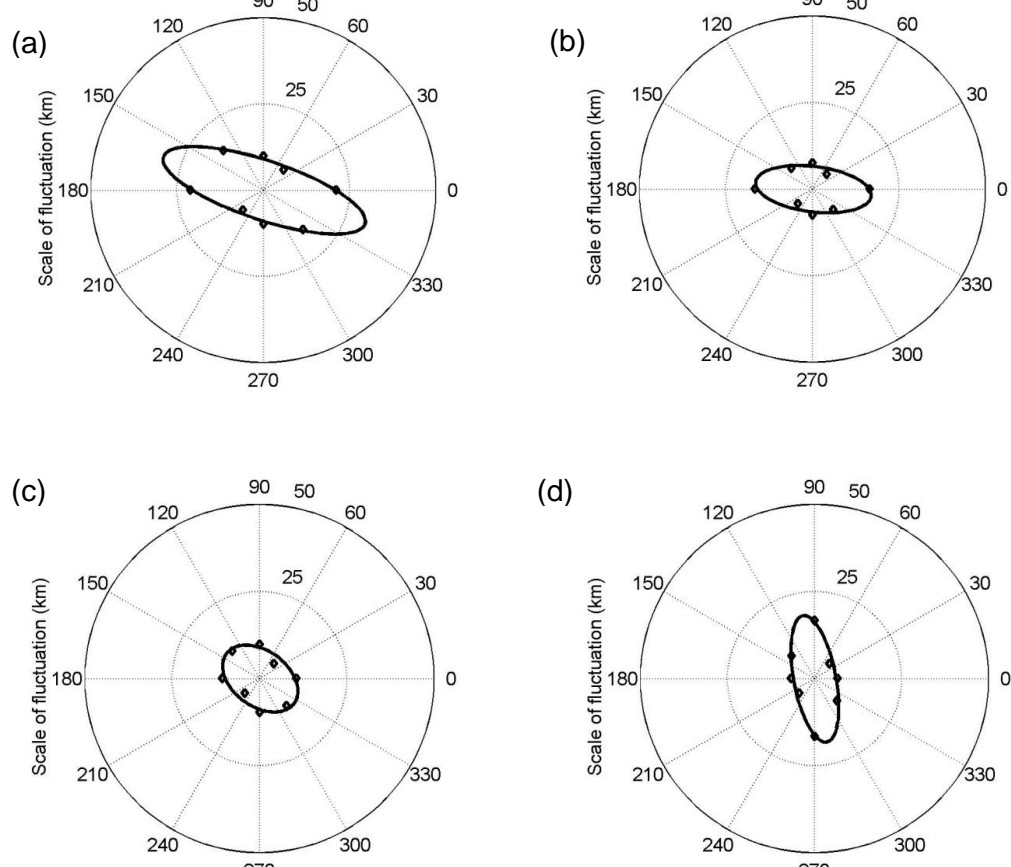

Figure 11. Scale of fluctuation values and ellipse-fitting curves for the 5-7 June 2008 storm: (a) maximum rolling 4-h rainfall, (b) maximum rolling 12-h rainfall; (c) maximum rolling 24-h rainfall; (d) maximum rolling 36-h rainfall.




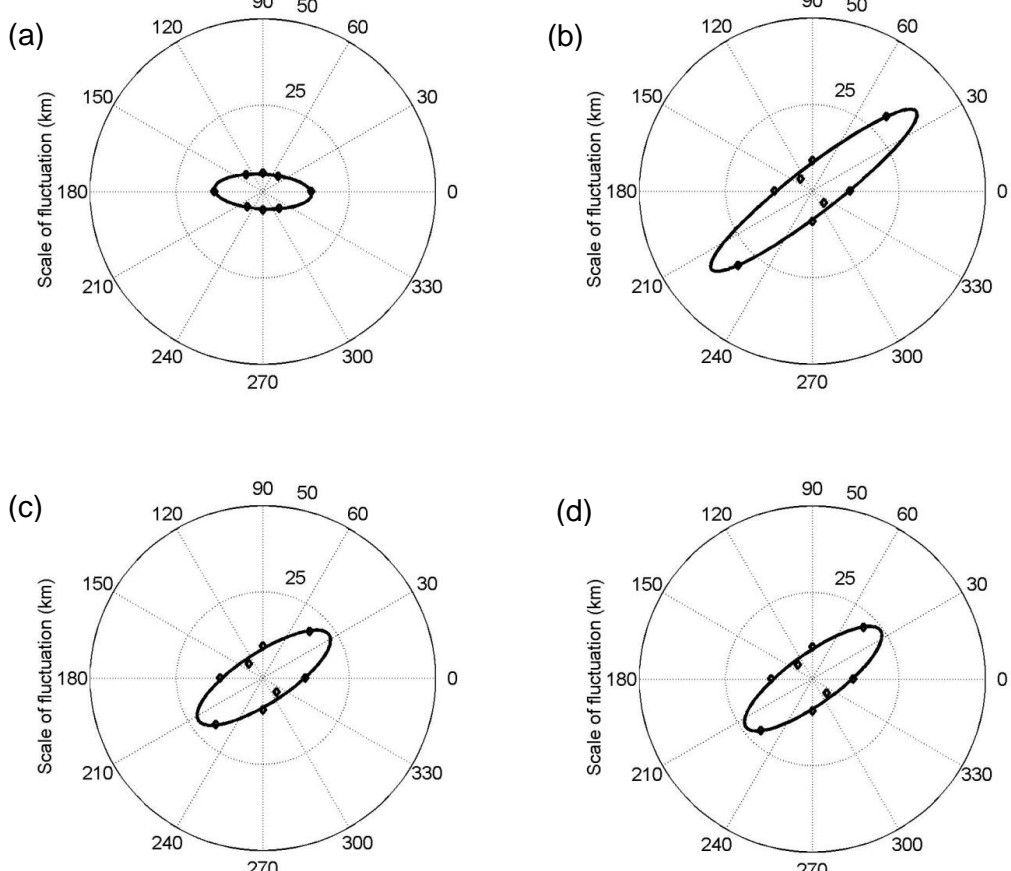

Figure 12. Scale of fluctuation values and ellipse-fitting curves for the 17-21 August 2005
storm: (a) maximum rolling 4-h rainfall; (b) maximum rolling 12-h rainfall; (c) maximum
rolling 24-h rainfall; (d) maximum rolling 36-h rainfall.




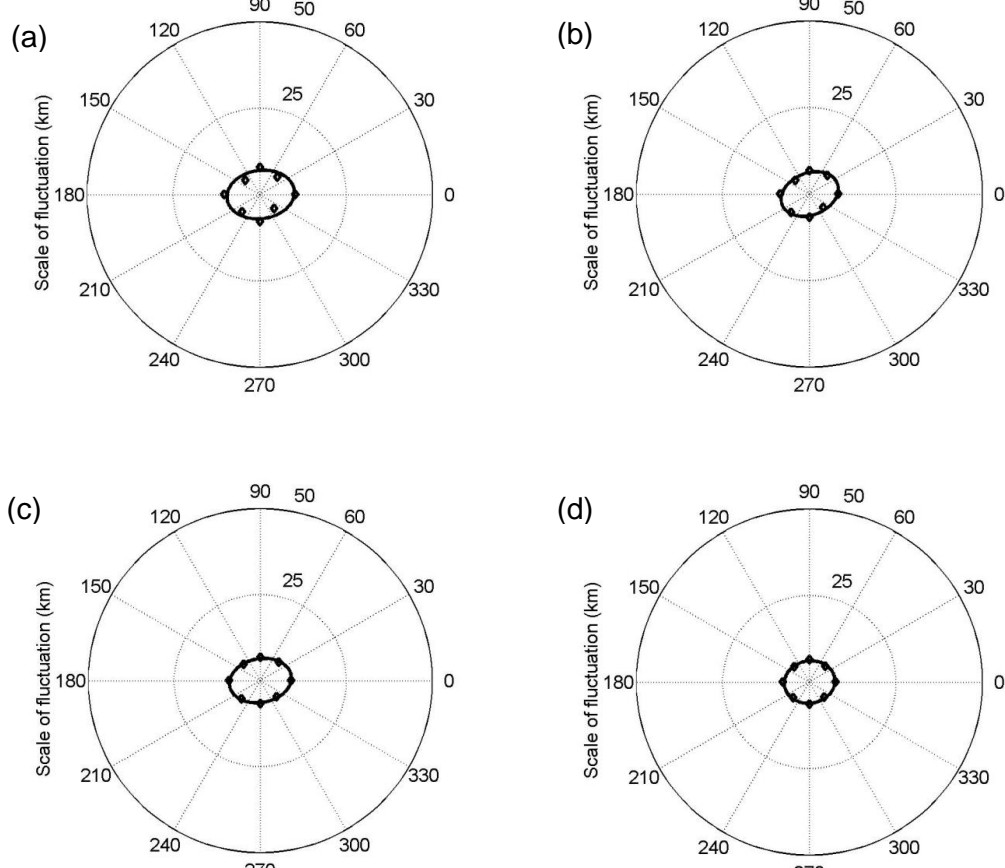

Figure 13. Scale of fluctuation values and ellipse-fitting curves for the 22-24 July 1994 storm:
(a) maximum rolling 4-h rainfall; (b) maximum rolling 12-h rainfall; (c) maximum rolling 24-h
rainfall; (b) maximum rolling 36-h rainfall.





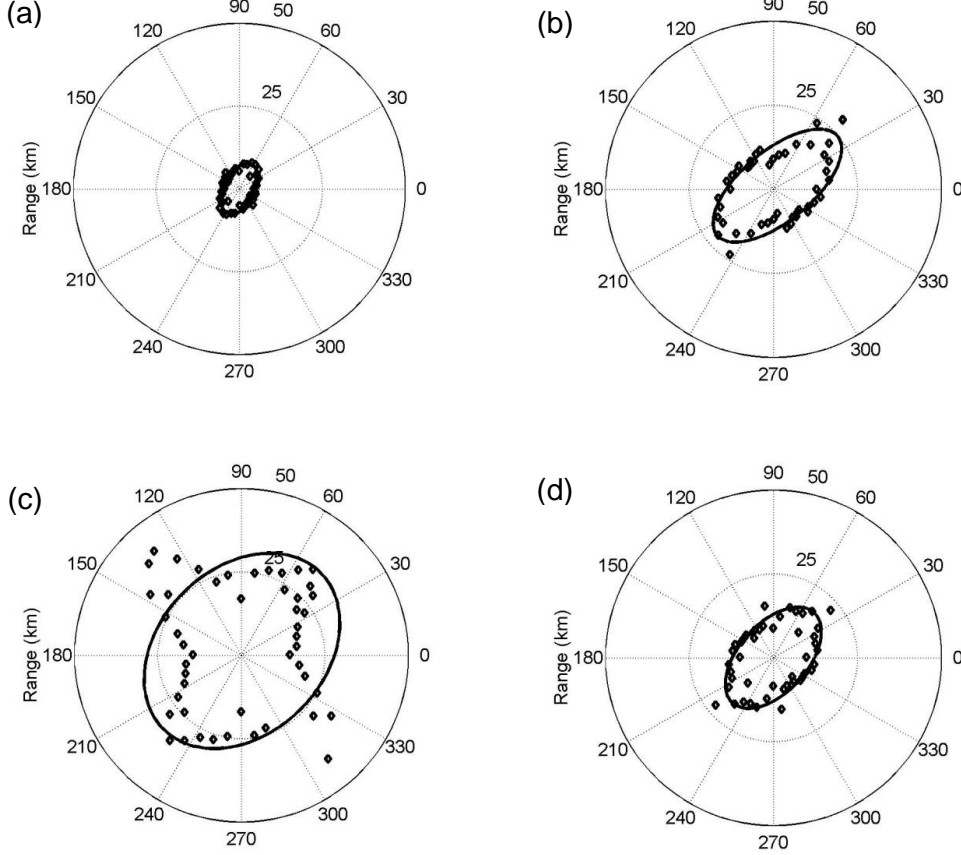

Figure 14. Range values for (a) the 18 May 2007 storm (16:30 pm); (b) the 19 May 2007 storm

(16:00 pm); (c) the 19 April 2008 storm (20:00 pm); (d) the 15 September 2009 storm (15:00

pm) (modified from Liu 2013).




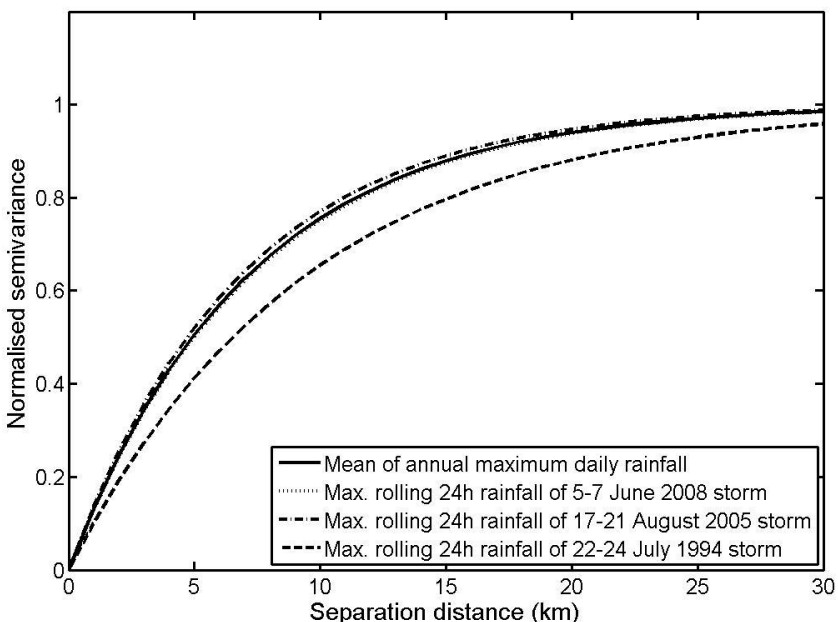

Figure 15. Normalised semivariances of the maximum rolling 24-hour rainfall of the three storms and the mean annual maximum daily rainfall in Hong Kong.