# Peer review of "Characterizing the spatial variations and correlations of large rainstorms for landslide study"

_Hydrology and Earth System Sciences, 2017_

## Referee Comment (RC1) · Anonymous Referee #1 · 27 Apr 2017

I consider that the manuscript can be accepted as is

---

## Referee Comment (RC2) · Anonymous Referee #2 · 1 Jun 2017

This paper presents a study related to the spatial characteristics of three large rainstorms in Hong Kong and aims to quantify their spatial correlation characteristics. The importance of this study is significant because such large rainstorms may trigger landslides and therefore their effects need to be considered when undertaking relevant landslide hazard analysis and risk management.

This Reviewer is a geotechnical engineer with experience in modelling landslides and associated coupled soil-water interactions, but with limited experience in the hydrological aspects of the problem.

The paper seems to present a thorough study of the spatial variations and correlations of large rainstorms and the study conducted is decent and worth for publication. The results are useful as an input in landslide hazard assessment. However, there is rather

limited connection between the spatial rainstorm variation and the potential for triggering a landslide. At the moment, the paper is a well presented study of the rainstorm that is perhaps poorly linked to the downstream application of landslide hazard analysis.

It is suggested that the Authors strengthen this relation by mentioning what other (e.g. geotechnical, environmental etc.) factors may ultimately affect the potential triggering of a landslide apart from rainfall intensity, e.g. slope inclination, rock/soil formations, vegetation, existence of civil infrastructure etc. Perhaps some examples of such factors may be added/reported from the studied area in Hong Kong.

Overall, the topic is relevant to HESS, the work is well-presented but there are a couple (additional to the technical issue discussed above) minor editorial issues that need to be addressed before the paper is accepted for publication:

1. Fig. 2 & 3: Is this from AECOM (2011) or AECOM & Lin (2015)? Apparently, the Reviewer cannot find the former citation in the Reference list.

2. Fig. 6: why the 1994 event shows larger rainfall mm values in 24h PMP than the 2008 event, whereas it shows smaller values in the 4h event?

---

## Author Comment (AC1) · 13 Jun 2017

We thank you so much for your positive recommendation.

———————————————

---

## Author Comment (AC2) · 13 Jun 2017

General Comments:

This paper presents a study related to the spatial characteristics of three large rainstorms in Hong Kong and aims to quantify their spatial correlation characteristics. The importance of this study is significant because such large rainstorms may trigger landslides and therefore their effects need to be considered when undertaking relevant landslide hazard analysis and risk management.

This Reviewer is a geotechnical engineer with experience in modelling landslides and associated coupled soil-water interactions, but with limited experience in the hydrological aspects of the problem. The paper seems to present a thorough study of the

spatial variations and correlations of large rainstorms and the study conducted is decent and worth for publication. The results are useful as an input in landslide hazard assessment. However, there is rather limited connection between the spatial rainstorm variation and the potential for triggering a landslide. At the moment, the paper is a well presented study of the rainstorm that is perhaps poorly linked to the downstream application of landslide hazard analysis. It is suggested that the Authors strengthen this relation by mentioning what other (e.g. geotechnical, environmental etc.) factors may ultimately affect the potential triggering of a landslide apart from rainfall intensity, e.g. slope inclination, rock/soil formations, vegetation, existence of civil infrastructure etc. Perhaps some examples of such factors may be added/reported from the studied area in Hong Kong.

Reply:

We thank this reviewer for the valuable comments and suggestions provided, which help improve the quality of the paper.

This study is expected to provide essential input for landslide risk assessment. Historical records show that the spatial rainstorm variation and potential for triggering landslides are closely correlated. The Geotechnical Engineering Office (GEO) compiled an inventory of historical natural-terrain landslides in Hong Kong in the mid-1990s, which was known as Natural Terrain Landslide Inventory (NTLI) (King 1999). Since then, GEO has enhanced it based on both high- and low-flight aerial photographs taken during 1924-2013 (Maunsell-Fugro Joint Venture 2007), with records of 19,763 natural terrain landslides and debris flows up to 2013 and 89,571 relict natural terrain landslides. We extracted the data of natural terrain landslides that occurred in three years (1994, 2005 and 2008), and plot the distributions of landslide volumes and the maximum 24-h rolling rainfall, as shown in Figure 1 in this reply. There is an obvious relation between the spatial rainstorm variation and occurrence of landslides. Characterizing the spatial characteristics of storms is therefore essential for assessing rainfall-triggered landslide hazards.

Numerical analyses have also been conducted to establish the relation between rainfall characteristics and landslides (e.g. Gao et al., 2015; Gao et al., 2016). Geotechnical and environmental factors, such as slope gradient, rock/soil formations, vegetation, and presence of civil infrastructure, are believed to ultimately affect the triggering of landslides apart from rainfall intensity. These factors (as shown in Fig. 2 in this reply) should be considered.

King, J. P.: Natural Terrain Landslide Study: Natural Terrain Landslide Inventory, GEO Report No. 74, Hong Kong: Geotechnical Engineering Office, HKSAR, 1999.

Maunsell-Fugro Joint Venture: Final report on compilation of the Enhanced Natural Terrain Landslide Inventory (ENTLI), Maunsell-Fugro Joint Venture & Geotechnical Engineering Office, HKSAR, 2007.

Gao, L., Zhang, L. M., Chen, H. X.: Likely scenarios of natural terrain shallow slope failures on Hong Kong Island under extreme storms, Natural Hazards Review, ASCE:B4015001, 2015.

Gao, L., Zhang L. M., Chen, H. X., Shen, P.: Simulating debris flow mobility in urban settings, Engineering Geology, 214:67-78, 2016.

Specific Comments:

Overall, the topic is relevant to HESS, the work is well-presented but there are a couple (additional to the technical issue discussed above) minor editorial issues that need to be addressed before the paper is accepted for publication:

1. Fig. 2 & 3: Is this from AECOM (2011) or AECOM & Lin (2015)? Apparently, the Reviewer cannot find the former citation in the Reference list.

2. Fig. 6: why the 1994 event shows larger rainfall mm values in 24h PMP than the 2008 event, whereas it shows smaller values in the 4h event?

Reply:

Thanks a lot for pointing out issues on the reference AECOM & Lin (2015). Figs. 2, 3 and 6 are modified from AECOM and Lin (2015). We have updated the references for the three figures: "Figure 2. 24-hour orographic intensification factors in Hong Kong (modified from AECOM and Lin (2015))"; "Figure 3. Generalized convergence component pattern with (a) NE-SW orientation 45° (b) ENE-WSW orientation 22.5° centred at Hong Kong Island; (c) NE-SW orientation 45° (d) ENE-WSW orientation 22.5° centred at Lantau Island; (e) NE-SW orientation 45° (f) ENE-WSW orientation 22.5° centred at Tai Mo Shan (modified from AECOM and Lin (2015)"; and "Figure 6. Magnitudes of the three storms characterized by (a) 4-h PMP, and (b) 24-h PMP (Modified from AECOM and Lin (2015))."

AECOM and Lin, B.: 24-hour probable maximum precipitation updating study, GEO Report 377 No. 314, Hong Kong: Geotechnical Engineering Office, HKSAR, 2015.

In Fig. 6, the magnitudes of the three rainstorms are described using depth-area curves, which are determined based on the spatial interpolation values of the maximum 4-h or 24-h rolling rainfall. According to the results in Table1, the 1994 storm event has a greater 24-h rainfall amount (956 mm) than the 2008 storm event (623 mm), but a smaller 4-h rainfall amount (365 mm) than the 2008 storm event (384 mm). As indicated in Table 1 and Fig. 4, the maximum 4-h rolling rainfall value was recorded in the 2008 event while the maximum 4-h rolling rainfall value was recorded in the 1994 event.

[Figure]

**Fig. 1.** Figure 1. Spatial distributions of the maximum 24-h rolling rainfall and the natural terrain landslides triggered in Hong Kong: (a) 7 June 2008 storm, (b) 20 August 2005 storm, (c) 23 July 1994 storm

[Figure]

**Fig. 2.** Figure 2. Geotechnical and environmental factors that affect triggering of landslides on western Hong Kong Island.

---

## Author Response (AR1)

**Hydrology and Earth System Science**

**Response to Review Comments**

Manuscript number: hess-2017-117 R1
Title: Characterizing the spatial variations and correlations of large rainstorms for landslide study
Article type: Research article
Iteration: Minor Revision
Authors: L. Gao, L. M. Zhang and M. Q. Lu

We would like to thank the two reviewers and the editor for their comments and constructive suggestions. We have considered these comments and revised the manuscript accordingly. Listed below please find our written responses to the reviewers' comments. Both the reviewers' comments and our responses are tabulated for ease of reference. The major changes are also highlighted in the text.

**Response to the Comments from Editor**

| Comments | Responses |
|---|---|
| The Editor has decided that minor revisions are necessary before the manuscript can be accepted. | Thank you so much for your kind recommendation. We have considered all the comments and revised the paper accordingly. |

**Response to the Comments from Reviewer #1**

| Comments | Responses |
|---|---|
| I consider that the manuscript can be accepted as is. | We thank you so much for your positive recommendation. |

**Response to the Comments from Reviewer #2**

| Comments | Responses |
|---|---|
| This paper presents a study related to the spatial characteristics of three large rainstorms in Hong Kong and aims to quantify their spatial correlation characteristics. The importance of this | We thank this reviewer for the valuable comments and suggestions provided, which help improve the quality of the paper. |

study is significant because such large rainstorms may trigger landslides and therefore their effects need to be considered when undertaking relevant landslide hazard analysis and risk management.

This Reviewer is a geotechnical engineer with experience in modelling landslides and associated coupled soil-water interactions, but with limited experience in the hydrological aspects of the problem.

| | |
|---|---|
| The paper seems to present a thorough study of the spatial variations and correlations of large rainstorms and the study conducted is decent and worth for publication. The results are useful as an input in landslide hazard assessment. However, there is rather limited connection between the spatial rainstorm variation and the potential for triggering a landslide. At the moment, the paper is a well presented study of the rainstorm that is perhaps poorly linked to the downstream application of landslide hazard analysis. It is suggested that the Authors strengthen this relation by mentioning what other (e.g. geotechnical, environmental etc.) factors may ultimately affect the potential triggering of a landslide apart from rainfall intensity, e.g. slope inclination, rock/soil formations, vegetation, existence of civil infrastructure etc. Perhaps some examples of such factors may be added/reported from the studied area in Hong Kong. | This study is expected to provide essential input for landslide risk assessment. In the revised paper, several sentences and two figures (Figs. 1 and 2) have been added to discuss the connection between the spatial rainstorm variation and the potential for triggering a landslide. (Lines 28-42, Page 2):

 "Historical records show that the spatial rainstorm variation and the potential for triggering landslides are closely correlated. The Geotechnical Engineering Office (GEO) maintains a Natural Terrain Landslide Inventory (NTLI) (King, 1999; Maunsell-Fugro Joint Venture, 2007), which has records of 19,763 natural terrain landslides and debris flows up to 2013 and 89,571 relict natural terrain landslides. The data of natural terrain landslides that occurred on 5-7 June 2008 are extracted and the distributions of the landslide volume and the maximum 24-h rolling rainfall are plotted in Fig. 1. There is a close correspondence between the observed landslide volume and the maximum 24-h rolling rainfall in space. Characterizing the spatial characteristics of storms is therefore essential for assessing rainfall-triggered landslide hazards.

 Numerical analyses have also been conducted to establish the relation between rainfall |

| | characteristics and landslides (e.g. Gao et al., 2015; Gao et al., 2016). Geotechnical and environmental factors, such as slope gradient, rock/soil formations, groundwater conditions, vegetation, and presence of civil infrastructure, are believed to ultimately affect the triggering of landslides in addition to rainfall intensity. The main factors that affect triggering of natural terrain landslides are summarised in Fig. 2." |
|---|---|
| Overall, the topic is relevant to HESS, the work is well-presented but there are a couple (additional to the technical issue discussed above) minor editorial issues that need to be addressed before the paper is accepted for publication: 1. Fig. 2 & 3: Is this from AECOM (2011) or AECOM & Lin (2015)? Apparently, the Reviewer cannot find the former citation in the Reference list. | Thanks a lot for pointing out issues on the reference AECOM & Lin (2015). Figs. 2, 3 and 6 (now Figs. 4, 5 and 8 in the revised version) are modified from AECOM and Lin (2015). We have updated the references for the three figures: "**Figure 4.** 24-hour orographic intensification factors in Hong Kong (modified from AECOM and Lin, 2015)." "**Figure 5.** The trend surfaces of 24-h PMP with (a) NE-SW orientation 45°; (b) ENE-WSW orientation 22.5° centred at Hong Kong Island; (c) NE-SW orientation 45°; (d) ENE-WSW orientation 22.5° centred at Lantau Island; (e) NE-SW orientation 45°; (f) ENE-WSW orientation 22.5° centred at Tai Mo Shan (modified from AECOM and Lin, 2015)." "**Figure 8.** Magnitudes of the three storms characterized by (a) 4-h PMP, and (b) 24-h PMP (modified from AECOM and Lin, 2015)." |
| 2. Fig. 6: why the 1994 event shows larger rainfall mm values in 24h PMP than the 2008 event, whereas it shows smaller values in the 4h event? | In Fig. 6 (now Fig. 8), the magnitudes of the three rainstorms are described using depth-area curves, which are determined based on the spatial interpolation values of the maximum 4-h or 24-h rolling rainfall. According to the results in Table 1, the 1994 storm event has a greater 24-h rainfall amount (956 mm) than the 2008 storm event (623 mm), but a smaller 4-h rainfall amount (365 mm) than the 2008 storm event (384 mm). |

| | As indicated in Table 1 and Fig. 8, the maximum 4-h rolling rainfall value was recorded in the 2008 event while the maximum 24-h rolling rainfall value was recorded in the 1994 event. We emphasize this observation in Lines 206-208, Page 9:

 "The 22-24 July 1994 storm is the largest among the three storms with regard to the amounts of the maximum rolling 1-h and 24-h rainfall. However, in terms of the maximum rolling 4-h rainfall, the 5-7 June 2008 storm is the most critical." |
|---|---|

**References**

King, J. P.: Natural Terrain Landslide Study: Natural Terrain Landslide Inventory. GEO Report No. 74, Geotechnical Engineering Office, Hong Kong, 1999.

Maunsell-Fugro Joint Venture: Final Report on Compilation of the Enhanced Natural Terrain Landslide Inventory (ENTLI). Maunsell-Fugro Joint Venture & Geotechnical Engineering Office, Hong Kong, 2007.

Gao, L., Zhang, L. M., Chen, H. X.: Likely scenarios of natural terrain shallow slope failures on Hong Kong Island under extreme storms, Natural Hazards Review, ASCE: B4015001, 2015.

Gao, L., Zhang L. M., Chen, H. X., Shen, P.: Simulating debris flow mobility in urban settings, Engineering Geology, 214:67-78, 2016.

AECOM and Lin, B.: 24-hour Probable Maximum Precipitation Updating Study. GEO Report 377 No. 314, Hong Kong: Geotechnical Engineering Office, Hong Kong, 2015.